# Preparation and Mechanical-Fatigue Properties of Elastic Polyurethane Concrete Composites

**DOI:** 10.3390/ma14143839

**Published:** 2021-07-09

**Authors:** Zhen Jia, Dongzhe Jia, Quansheng Sun, Yanqi Wang, Hongjian Ding

**Affiliations:** 1School of Civil Engineering, Harbin University, Harbin 150040, China; zhenzhen7826@163.com; 2School of Civil Engineering, Northeast Forestry University, Harbin 150040, China; jdz724106875@126.com (D.J.); wangyanqi0514@163.com (Y.W.); dinghj@nefu.edu.cn (H.D.)

**Keywords:** elastic polyurethane cement (EPUC), rubber particles, constitutive relation, microscopic morphology, fatigue prediction

## Abstract

In order to solve issues related to bridge girders, expansion devices and road surfaces, as well as other structures that are prone to fatigue failure, a kind of fatigue-resistant elastic polyurethane concrete (EPUC) was obtained by adding waste rubber particles (40 mesh with 10% fine aggregate volume replacement rate) to conventional engineering polyurethane concrete (PUC). Based on the preparation and properties of EPUC, its constitutive relation was proposed through compression and tensile tests; then, a scanning electron microscope (SEM), an atomic force microscope (AFM) and a 3D non-contact surface profilometer were used to study the failure morphology and micromechanisms of EPUC. On this basis, four-point bending fatigue tests of EPUC were carried out at different temperature levels (−20 °C, 0 °C, 20 °C) and different strain levels (400 με~1200 με). These were used to analyze the stiffness modulus, hysteresis angle and dissipated energy of EPUC, and our results outline the fatigue life prediction models of EPUC at different temperatures. The results show that the addition of rubber particles fills the interior of EPUC with tiny elastic structures and effectively optimizes the interface bonding between aggregate and polyurethane. In addition, EPUC has good mechanical properties and excellent fatigue resistance; the fatigue life of EPUC at a room temperature of 600 με can grow by more than two million times, and it also has a longer service life and reduced disease frequency, as well as fewer maintenance requirements. This paper will provide a theoretical and design basis for the fatigue resistance design and engineering application of building materials. Meanwhile, the new EPUC material has broad application potential in terms of roads, bridges and green buildings.

## 1. Introduction

Due to rapid developments in the global economy, significant increases in traffic pressure and changes in inclement weather conditions, roads and bridges have become more prone to fatigue damage. Conventional materials such as concrete are characterized by high stiffness, and they will struggle to resist the large deformations brought about by impact loads in the future. As such, it vital that we find an elastic material with a short maintenance time, light weight, high strength, a strong cohesive force and excellent structural integrity for application in the field of road and bridge construction and repair. Importantly, the annual number of scrap tires has reached 1.5 billion globally. Like plastic products, waste tires are difficult to degrade and do not disappear naturally after being buried underground for decades, causing a phenomenon known as “black pollution”. Therefore, the effective recovery and reuse of waste rubber to reduce environmental pollution can make important contributions to the construction of an eco-friendly society [1]. EPUC (EPUC) is a composite material mixed with one component or two component Polyurethane as the binder, cement sand as the filler aggregate and mixed with a certain proportion of rubber particles. Compared with conventional concrete, EPUC has lower density, higher strength and a shorter fitting time and, in the flow state, it is a kind of colloid in nature, with strong bonding force, no peeling damage and a strong ability to limit cracks; however, it has high usage costs and considerable construction process requirements [2,3]. At the same time, the addition of rubber particles can effectively reduce the internal stress and failure cracks of EPUC; it effectively inherits the energy absorption and impact resistance characteristics of the rubber particles [4], which can protect engineering structures and resist fatigue, thus prolonging a given structure’s service life.

Rubber concrete and polyurethane concrete have attracted many researchers because of their excellent physical and mechanical properties. Rubber concrete can effectively improve the elastic deformation capacity of concrete, as well as its heat insulation, frost resistance, shock absorption and sound insulation effects. Eldin and Senouci, as well as Khatib et al. [5,6], selected rubber particles with the same diameter as coarse aggregate and mixed them at a certain aggregate volume substitution rate. The test results show that the compressive strength and splitting tensile strength of rubber concrete decrease with the increase in rubber particle content, but rubber can nevertheless significantly enhance the concrete’s energy absorption capacity. Polyurethane concrete has been widely used in engineering due to its excellent mechanical properties such as tensile and compression strength, elongation and quick molding and curing [7,8,9,10]. Haleem K. Hussain et al. [11,12] carried out mechanical tests on the flexural and compressive strength of polyurethane concrete composites, and they obtained appropriate tensile and compressive strength under different mix ratios of polyurethane concrete. On this basis, reinforcement and re-failure tests were carried out on T beams with different damage degrees. The results show that a polyurethane concrete-reinforced bridge can significantly improve the ultimate load of the original beam and reduce structural cracks. Leon Agavriloaie et al. [13] prepared a new type of polymer concrete with epoxy polyurethane acrylate and aggregate as raw materials, and tested its properties through bending tensile strength, elastic modulus, drawing stress and bonding stress. Its thermal and physical properties were studied, including bulk density, humidity, thermal conductivity, linear thermal expansion, thermal impact strength, chemical corrosion resistance, freezing and thawing resistance and water absorption. The test results show that this epoxy polyurethane acrylate concrete composite material has excellent mechanical properties, a light weight and high strength, and it has the potential to replace traditional building materials. Zun-xiang et al. [14] diluted one-component polyurethane with acetone to study its viscosity, dispersion area, surface tension and bonding strength. The diluted polyurethane was encapsulated in a quartz glass tube and its compatibility with concrete and its survival rate were analyzed. The interface between polyurethane and concrete was studied by scanning electron microscopy, while the elemental composition of polyurethane was determined by energy dispersive X-ray energy spectrometry; the mechanical properties and self-healing efficiency of the polyurethane self-healing agent were determined. The results showed that the glass capsule containing polyurethane healing agent that was embedded in the concrete could make use of its self-healing properties. Iman Kattoof Harith [15] prepared cube, cylinder and prism samples by different curing methods (water, wet, film curing compound sealing and air curing), and studied compressive strength, elastic modulus and drying shrinkage at different ages. The research results show that the wet-cured specimens have the highest compressive strength at various ages, indicating the potential of polyurethane foam concrete in structural application. Shigang Ai et al. [16] experimented on the dynamic compressive properties of polyurethane polymer concrete (PPC) by using the Hopkinson split bar (SHPB) system; they established a two-dimensional finite element model of the SHPB system, including a heterogeneous mesoscale model of the PPC sample. The explicit numerical method based on LS-DYNA coding was used to simulate the compressed PPC specimens under different strain rates, and the results show that PPC materials have excellent dynamic compression characteristics. Jun Chen et al. [17] analyzed the differences between polyurethane concrete pavements and asphalt concrete pavements in terms of ice coverage, as well as their respective effects on strength and shear strength. The results showed that polyurethane concrete possesses excellent de-icing attributes. The findings from this study provide a basis for the notion that polyurethane concrete can be applied to roads in cold areas as a means of improving traffic volume and safety in winter. Bjorn Van Belleghem et al. [18] used self-healing polyurethane concrete with low viscosity to strengthen cracks in bridge structures and prevent the rapid expansion of cracks from causing corrosive substances to enter the steel bars, thus significantly reducing their corrosion rate. In summary, there is much research examining the detailed ratio, constitutive relationship, mechanical properties and specific applications of rubberized concrete and conventional PUC materials, but there is minimal research regarding the combination of rubberized concrete and conventional PUC materials.

In this study, we combine the excellent properties of rubberized concrete and conventional PUC materials to systematically study the preparation, mechanics and fatigue properties of EPUC. We hope that the findings from this process can effectively solve the problem of fatigue damage in building materials as it currently stands. As such, we provide a means by which to effectively recover and reuse waste rubber, reduce environmental pollution and make an important contribution to the construction of an eco-friendly society. Moreover, our study has significant potential regarding its application in the reinforcement and repair of existing structures and green buildings.

## 2. Experimental Methods

### 2.1. Materials

Polyphenylene isocyanate (MDI, industrial-grade) was purchased from Beste Chemical Technology Co. Ltd., Dalian, China. Combined polyether (ES305, industrial-grade) was purchased from Jiangsu Hai’An Petrochemical Co. Ltd., Haian, China. The catalyst (phosphoric acid) was purchased from Shandong Yisheng Polyurethane Co. Ltd., Jinan, China. Portland cement (average particle size = 1.455 μm) was purchased from Harbin Honggu Cement Manufacturing Co. Ltd., Harbin, China. Rubber particles (10 mesh, 40 mesh and 80 mesh) were made by grinding used rubber tires. Aggregate gradation adopted AC-10 fine-grained concrete typical gradation, and the specific particle size and ratio are shown in Table 1. All chemicals were used as received without further purification.

### 2.2. Preparation of Reversible Photochromic PUC

The preparation process of EPUC improved on the basis of previous studies, and it is shown in Figure 1 [19]. Figure 2 shows the content of EPUC. First, the waste rubber tires were polished into particles of 40 mesh, and the waste rubber particles were pretreated. In this paper, more conventional water was selected as the pretreatment method for the test. Then, cement, aggregate and rubber particles were dried at 100~110 °C for 2 h (101-2a electrothermal blast drying oven); during this period, the free water in the concrete was fully removed by turning every 20 min. Subsequently, the isocyanate was mixed with the composite polyether at a ratio of 1:1. After stirring evenly for 2 min, the dried aggregate and rubber particles (40 mesh with 10% fine aggregate volume replacement rate) were added and stirred until evenly dispersed, and the catalyst was added according to 1% of the content of polyurethane. When the temperature rose and the mixture gradually became colloidal from liquid, it was poured into the mold coated with demolding agent. When the specimen was demolded 24 h after curing, its mechanical strength could reach more than 80% of the final strength.

### 2.3. Mechanical Properties Test

(1) Compression test: The cube compressive strength test of EPUC was carried out at a room temperature of 20 °C, and the size of the sample was a standard cube of 100 mm × 100 mm × 100 mm [20]. In order to reduce the ferrule effect caused by the friction between the upper and lower steel plates of the universal testing machine and the specimen, Vaseline was applied to the compression surface of the specimen, and horizontal strain gauges and vertical strain gauges were affixed to the free surface of the EPUC to test the Poisson’s ratio. Figure 3 shows the JM3812 multi-functional static strain acquisition system (Jingming Technology Co. Ltd., Yangzhou, China), and the loading speed of the universal testing machine (Sansizongheng Technology Co. Ltd., Shenzhen, China) was 0.5 mm/min [21]. Figure 4 shows the compressive strength test diagram of EPUC.

The compressive strength of EPUC is calculated as follows:(1)$$f_{ca}=\frac{F}{A}$$
where: $f_{ca}$—compressive strength of concrete cube specimens;

*F*—ultimate load (N);*A*—compression area (mm^2^).

(2) Straight-tension test: In order to eliminate the uncertain influence of a single specimen, six sets of parallel specimens (ZL01~ZL06) were selected for the straight-pull test. A dumbbell-type sheet was used as the standard specimen [22] of the straight-tension test. The thickness of the specimen was 12.7 mm, the width of the thinnest part in the middle was 30.0 mm and the width of both sides was 60.0 mm; the specific size is shown in Figure 5. The test fixture was self-designed with ball hinges above the fixture to ensure that the specimen could bear a vertical load. The tensile test fixture was installed on the universal test machine, and then the dumbbell specimen was put into the fixture for the straight-pull test at a tensile speed of 50.0 N/s. The resistance strain gauge was pasted on the side of the specimen, and the strain data of the specimen were collected by a JM3812 dynamic strain acquisition instrument. The straight tensile test of EPUC is shown in Figure 6.

### 2.4. Fatigue Test

Based on the research results for the SHRPA-003A Strategic Highway [23], and in view of the characteristics of bending deformation and tensile fatigue in the anchorage area of the expansion joint, a four-point bending fatigue test was carried out on EPUC under the coupling action of temperature and mechanics. The testing principle and loading mode of the four-point bending test are shown in Figure 7.

The fatigue test instrument in this paper was a UTM-30 fatigue-loading test system (IPC Global Co. Ltd., Sydney, Australia) with a limit load of 30 kN UTM-30 (Figure 8). Three temperature gradients—room temperature 20 °C, critical temperature 0 °C and winter temperature −20 °C—were used in this experiment. Three groups of parallel specimens were selected for each stress level, and the “abandonment method” in the probability and statistics method was applied for evaluation. When the difference between the test value of a group and the mean value was 1.15 times greater than the standard deviation, the number of tests was discarded and supplemented again until the test passed.

The main parameters such as loading frequency, loading waveform and strain level need to be calibrated before the four-point bending fatigue test, as different parameter choices will greatly affect the fatigue life of EPUC. In this paper, a loading frequency of 10 Hz and a continuous uninterrupted semi-positive vector wave were selected for the fatigue test [24]; the strain levels were 400 με, 600 με, 800 με, 1000 με and 1200 με [25]. When the bending stiffness modulus of the specimen was reduced to half of the initial stiffness modulus, the experiment was stopped, and the specimen was considered to have been damaged by fatigue. When the load accumulative action times were exceeded by 2 × 10^6^ times and still failed to meet the above conditions, the test was stopped, and the fatigue life of the specimen under these conditions was deemed to be infinite.

The maximum tensile stress ($\sigma_{t}$) of the specimen is calculated according to Equation (2) and maximum tensile strain ($\varepsilon_{t}$) is calculated according to Equation (3).
(2)$$\sigma_{t}=\frac{L\times P}{\omega \times h^{2}}$$
(3)$$\varepsilon_{t}=\frac{12\times \delta \times h}{3\times L^{2}-4\times a^{2}}$$
where *L* = the span of bending specimen (m);

*P* = the peak load (N); ω = the width of the bending specimen;*h* = the height of the bending specimen;δ = the maximum strain of the specimen center;*a* = the intermediate spacing between adjacent collets (m).

The stiffness modulus is calculated according to Equation (4), the lag Angle is calculated according to Equation (5), the dissipated energy within the unit load cycle is calculated according to Equation (6) and the cumulative dissipated energy is calculated according to Equation (7).
(4)$$S=\frac{\sigma_{t}}{\varepsilon_{t}}$$
where *S* = modulus of bending stiffness (Pa)
(5)φ=360×f×t
where φ = lag Angle (°);

*f =* loading frequency (Hz);*t =* time of hysteresis deformation (s).
(6)$$E_{D}=\pi \times \sigma_{t}\times \varepsilon_{t}\times \sin\varphi$$
where $E_{D}$ = dissipated energy of a single cycle (J/m^3^).
(7)$$E_{CD}=\overset{n}{\underset{i=1}{\sum }}E_{CDi}$$
where $E_{CD}$ = cumulative dissipated energy (J/m^3^);$E_{CDi}$ = dissipated energy of a single cycle (J/m^3^).

### 2.5. Characterizations

A scanning electron microscope (Carl Zeiss Management Co. Ltd., Shanghai, China) was used to observe the surface and section morphology of EPUC with the optimal ratio. Atomic force microscopy (Leimai Technology Co. Ltd., Hangzhou, China) (tap mode, scanning area = 300 × 300 nm^2^) and 3D non-contact surface profilometry(Bruker Technology Co. Ltd., Beijing, China) were used to investigate the surface morphology and roughness (three points were selected for each sample; the focus moving synthesis method, which is used for samples with concave and convex textures, was used).

## 3. Results and Discussion

### 3.1. Mechanical Properties

(1)Compression properties

In order to eliminate the uncertain influence of a single specimen, six sets of parallel specimens (F01~F06) were selected to test compressive strength and density. The compressive strength of EPUC is shown in Table 2. The compressive strength of EPUC is 58.5 MPa~61.2 MPa, while the average density is 1768.5 kg/m^3^ and the average compressive strength is 59.7 MPa. By comparison, it is found that, while EPUC meets the strength level of C50 concrete, its density is only 70% [26].

The average compressive strength of the six groups of specimens was taken to draw the stress–strain curve, as shown in Figure 9. The transverse–longitudinal strain test results for the bidirectional strain curve under compression are shown in Figure 10.

The mechanical properties of EPUC are similar to ordinary concrete, and the compression process can be divided into the following two stages. First is the elastic stage (3500 με~25,387.33 με), where the stress–strain curve of the specimen at the initial stage of loading is in positive linear correlation. The maximum compressive strength at this stage is 53.26 MPa, which is defined as the elastic limit of the material, and the strain at this stage is 25,387.33 με. Second is the yield stage (≥25,387.33 με), where the stress–strain curve of the specimen at the late loading period is approximately a downward parabolic curve with an ultimate compressive stress of 59.7 MPa and a failure strain of 35,665.72 με. After removing the fluctuation value of the initial test (the microstrain is between 0–3500 με), the stress–strain curves of the remaining two stages are fitted. The fitting equations of the two stages are as follows:3500 < *ε* < 25,387.33: *σ* = 2550.88*ε* − 11.1076, R^2^ = 0.99407
*ε* > 25,387.33: *σ* = 7356.98*ε* − 140.12*ε*^2^ + 7328.88*ε*^3^ − 55.0565, R^2^ = 0.99791

According to Hooke’s law of tension and compression (σ = *E*ε), stress and strain in elastic compression stage are proportional. *E* is the slope of the straight-line segment in Figure 9, and its value is approximately equal to the elastic modulus of EPUC (*E* = 2550.88 MPa); this value is much lower than the elastic modulus of ordinary concrete, which indicates that EPUC has excellent elastic deformation ability.

The slope in the linear elastic stage is the Poisson’s ratio of EPUC which, as can be seen from Figure 10, is about 0.29. Under the same vertical strain condition, the transverse shape variable of EPUC is larger than that of ordinary concrete, which indicates that it has a better elastic deformation ability.

(2)Tensile properties

The tensile strength of six groups of parallel specimens (ZL01~ZL06) is shown in Table 3, and the average tensile stress–strain curve under the optimal mix ratio of EPUC is obtained by taking the average value, as shown in Figure 11.

By fitting the stress and strain data of straight-tension specimens, the straight-tension stress–strain curve of EPUC is obtained. It can be seen from Figure 11 that the maximum tensile stress of EPUC in the elastic stage is 41.1 MPa, and the maximum tensile strain is 14,436.47 με. The results of tensile tests show that EPUC is an elastic material with excellent tensile properties greater than general C50 concrete. The fitting formula is as follows:*σ* = 2723.75*ε* − 14.2449, R^2^ = 0.9985

### 3.2. Analysis of Microscopic Morphology and Failure Mode

(1)Microscopic morphology analysis

Figure 12a–c shows that rubber particles are evenly distributed in EPUC. The content of rubber particles with a 10% volume replacement rate is moderate, so EPUC can maintain good elasticity while not excessively occupying the aggregate space, thus ensuring its strength. Additionally, 40 mesh rubber particles occupy a larger specific surface area and show a strong adsorption ability. Polyurethane cementing material provides a strong binding force for the aggregate and rubber particles, and can effectively fill the void in EPUC, making the internal void structure of EPUC material more detailed and compact.

As can be seen in Figure 13a,b, the surface of EPUC is rough and accompanied by small bubbles and particles. The small bubbles are caused by the gas released in the reaction process of isocyanate and composite polyether, while the small particles are rubber particles and fine aggregate. The sample morphology shown in Figure 14a,b is composed of geometric surface point clouds, and each point expresses its depth distance information to the horizontal surface through different colors. The colors are mainly classified into red, green and blue. The three colors on the surface are distributed alternately, and the surfaces of some concave and convex areas are rough. The above micro-morphology analysis results show that rubber particles bond well with EPUC and effectively fill the internal gaps of concrete to ensure a certain strength. Moreover, EPUC has a rough surface, which provides some friction and a certain wear resistance; in turn, this leads to it having a good road performance.

(2)Failure mode analysis

Three groups of specimens with different bone–glue ratios (ST-1~ST9) were selected for analysis to explore the influence of bone–cement ratio on the compressive failure morphology of EPUC. The axial compressive failure morphology of ST-1~ST-9 is shown in Figure 15.

ST1~ST6 is EPUC with a 10~15% bone–glue ratio. It has low strength and its failure pattern is irregular. The inside of the specimen is relatively loose, and various constituent materials do not play a full role. For specimens with a 20% bone–glue ratio of ST7~ST9, their failure mode is a typical X-type failure mode. Compared with ordinary concrete, there is no brittle crack or splitting damage, and only some small cracks exist; the morphological integrity of the damaged specimens is therefore intact. With the addition of rubber particles, the EPUC is filled with tiny elastic structures, and the interface bonding between aggregate and polyurethane is effectively optimized. The rubber particles evenly distributed in the concrete use their own elastic absorption of impact energy; they also slow down the internal impact of micro-cracks and significantly enhance the elastic deformation ability of EPUC so as to effectively slow down the further expansion of micro-cracks in the concrete and accelerate the destruction process of the structure.

The bone–glue ratio greatly affects the compressive strength of EPUC. When the bone–binder ratio content is low, there is a large gap between aggregates, and the bonding force between polyurethane cementing materials is weak. Rubber particles do not fully function, forming tiny “holes” in the concrete with no carrying capacity, resulting in a significant reduction in overall strength. The best bone–glue ratio is 20%, and polyurethane elastomer can completely wrap the aggregate, thus rendering the whole specimen uniform and smooth and negating the need for vibration compaction. Continuously increasing the bone–glue ratio has no obvious effect on strength, and it increases the difficulty of controlling water content during the test, thus affecting the forming state of EPUC.

### 3.3. Fatigue Properties

(1)Analysis of bending stiffness modulus

The bending stiffness modulus formula is shown in Equation (4).

From Figure 16, the bending stiffness of EPUC modulus attenuation comprises two main stages. The first stage is 10^5^ times before the fatigue cycle begins a slow linear downward trend; at this stage in the accumulation of fatigue damage, there are tiny cracks in the elastic polyurethane and concrete, but their mechanical properties are still considered at a high level. In the second stage, the accumulated damage to EPUC reaches a certain range after 100,000 fatigue cycles, and the bending stiffness modulus rapidly decays to less than 50% of the initial modulus in terms of bending tensile strength; this indicates that the material has fatigue damage. Both the low temperature and the high strain advance the second stage decay of the modulus in relation to bending stiffness.

(2)Analysis of initial stiffness modulus

The initial stiffness modulus of EPUC is an important characteristic of its fatigue curve and an important index to evaluate the fatigue life of EPUC. The initial stiffness modulus of EPUC fluctuates significantly. In order to eliminate the influence of unstable data in the early stage of the test, at least according to the AASHTO TP8 specification, the initial stiffness modulus of this test is the value of the rigidity modulus after the 50th fatigue cycle. The test results are shown in Figure 17.

At the same temperature, the initial stiffness modulus of EPUC decreases with the increase in strain. Softening at high temperature and hardening at low temperature at the same strain causes the initial stiffness modulus to decrease with the increase in temperature, but it is easy to advance the brittle failure time of specimens and shorten their fatigue life.

(3)Analysis of lag Angle

The lag angle is an important index for evaluating the viscoelastic and fatigue resistance properties of materials. The smaller the lag angle, the more elastic the material property tends to be; otherwise, it tends to be viscous. The lag angle in this paper is calculated according to Equation (5). EPUC is a kind of viscoelastic material with a hysteresis angle between 0~90°. Figure 18 shows the relationship between the lag angle of EPUC and the number of fatigue cycles at different temperatures and strain levels.

The higher the temperature, the higher the lag angle. This is because polyurethane material is a kind of heat-resistant cementifying material that displays high temperature softening behavior, in the same manner as asphalt, and its hardness decreases with the increase in temperature. At −20 °C, the lag angle of EPUC is less than 10, which is a perfect elastic material with good adaptability to the anchorage area of expansion joints in cold areas. At the same temperature, the lag angle of EPUC is proportional to the fatigue strain level, and the lag angle proves the ability of EPUC to resist fatigue failure from the side. The relationship between the hysteresis angle and the number of fatigue actions is an approximate downward parabola; the higher the strain level, the steeper the curve. It can be seen from Figure 18 that the hysteresis angle under a strain level of 400–600 με at 20 °C does not change much, showing a linear segment rising slightly and indicating that the hysteresis angle of EPUC under this condition is less affected by the number of fatigue cycles.

(4)Analysis of dissipated energy

Dissipated energy is the energy required by a material to resist external fatigue load. Under fatigue load, deflection and reorganization will occur inside the structure; important in this regard is the movement of aggregates such as sand and gravel, and the generation of new surfaces. The dissipated energy in a single load cycle is calculated according to Equation (6) and the cumulative dissipated energy is calculated by Equation (7). The variation trend in dissipated energy with fatigue times is shown in Figure 19.

The analysis shows that the energy consumption of EPUC decreases with an increase in loading times. The higher the strain level at the same temperature, the greater the energy consumed by the internal recombination of EPUC to resist external deformation. With the increase in temperature, the material tends to be viscous, and the dissipated energy at the same strain level decreases accordingly. The maximum dissipation energy is 0.95 kJ/m^3^ at −20 °C and 1200 με. The change in dissipated energy presents the following two-stage change trend. At the beginning of the fatigue cycle, the dissipated energy decreases rapidly due to the change in the internal structure of the material. When the number of cycles reaches more than 100,000, the dissipated energy gradually stabilizes and steadily decreases within a certain range.

(5)Fatigue equation of EPUC

The fatigue life *N* and strain level *ε* of EPUC at different temperatures are fitted logarithmically, and the results are shown in Figure 20. The higher the temperature, the longer the fatigue life of EPUC; the fatigue life of EPUC is more than two million times longer under normal temperature and the 600 με condition. From the correlation coefficient R2, as well as the fitting curve in the figure, it can be seen that the fatigue life equation of EPUC fits appropriately with the four-point bending fatigue test data.

According to the expression of Hooke’s law—*σ* = *E ε*—the *ε*-*N* fatigue equation is transformed into a classical *S-N* fatigue equation, as shown in Table 4 below:

We compared EPUC to the PUC fatigue equation proposed by Gao Hongshuai [25] and drew it in Figure 21 to explore the differences between them. It can be seen from Figure 21 that the fatigue life of EPUC increases at different temperatures. The increase is the largest at low temperatures, indicating that the addition of rubber particles to PUC can improve its fatigue life to a certain extent.

## 4. Conclusions

In this study, the preparation of EPUC and its mechanical properties was examined. We used a four-point bending fatigue test to present the *S-N* curve of the fatigue life of EPUC under the coupling actions of temperature and external force. Our findings are as follows:

(1) The best mixture ratio for EPUC is a cement-to-bone ratio of 20% and 40 mesh rubber particles mixed with a 10% fine aggregate volume substitution ratio. The specimen had an exemplary failure mode and considerable integrity, and only a few small cracks were apparent. In addition, the micro-morphology analysis results show that the rubber particles were evenly distributed in the EPUC and bonded well with the polyurethane material, which effectively guaranteed the strength of the elastic polyurethane concrete and showed that EPUC had a reliable enough surface roughness to meet road performance.

(2) The results of mechanical tests showed that the elastic limit of EPUC was 53.26 MPa and the maximum elastic strain was 25,387.33 με. The ultimate tensile stress was 41.1 MPa and the maximum tensile strain was 14,436.47 με. These results demonstrate that EPUC has excellent mechanical properties that are even better than general C50 concrete.

(3) At room temperature and low strain level, EPC has outstanding fatigue properties. The higher the strain level, the shorter the fatigue life of elastic polyurethane concrete; the fatigue life of EPUC at 400 με could grow by more than two million times. Taking 20 °C as the benchmark, it is clear that the lower the temperature, the shorter the fatigue life of EPUC. Therefore, adverse climatic conditions in cold areas will have an adverse effect on its application, but its fatigue life is still better than that of ordinary building materials.

## Figures and Tables

**Figure 1 materials-14-03839-f001:**
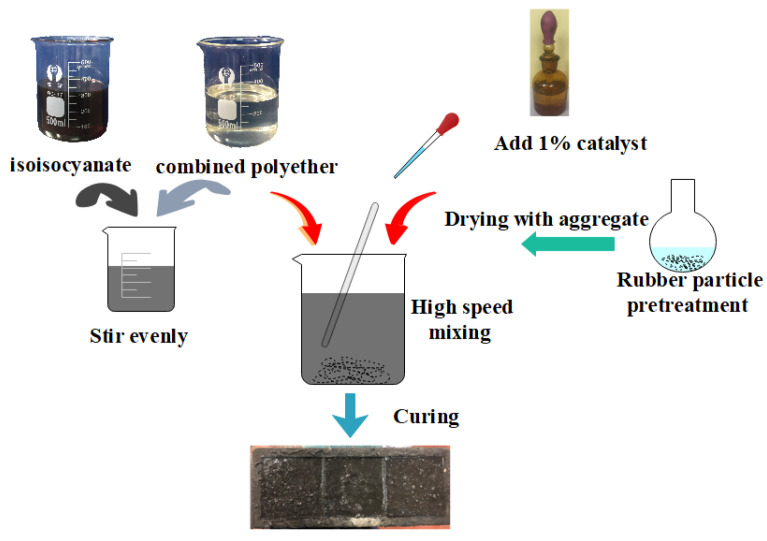
Preparation of the EPUC.

**Figure 2 materials-14-03839-f002:**
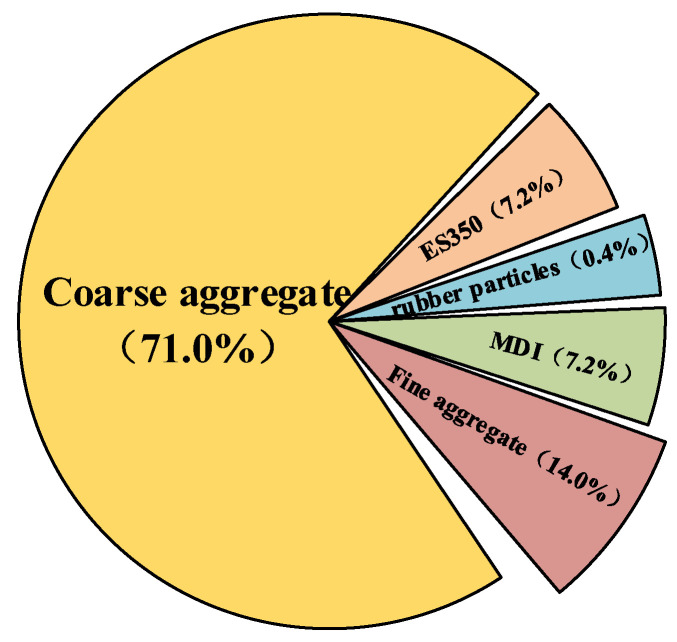
Content of EPUC.

**Figure 3 materials-14-03839-f003:**
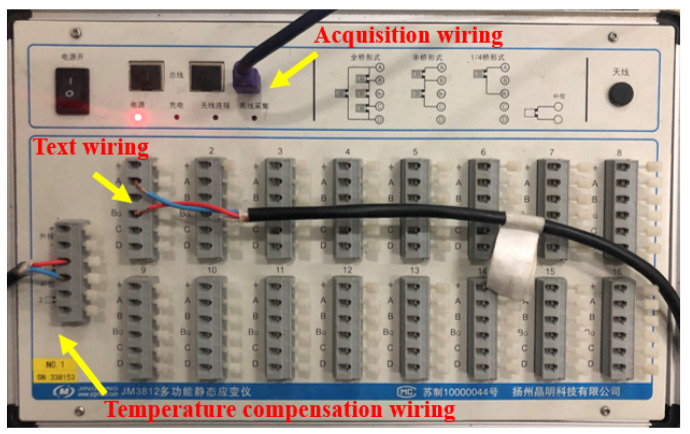
JM3812 Multi-functional static strain gauge.

**Figure 4 materials-14-03839-f004:**
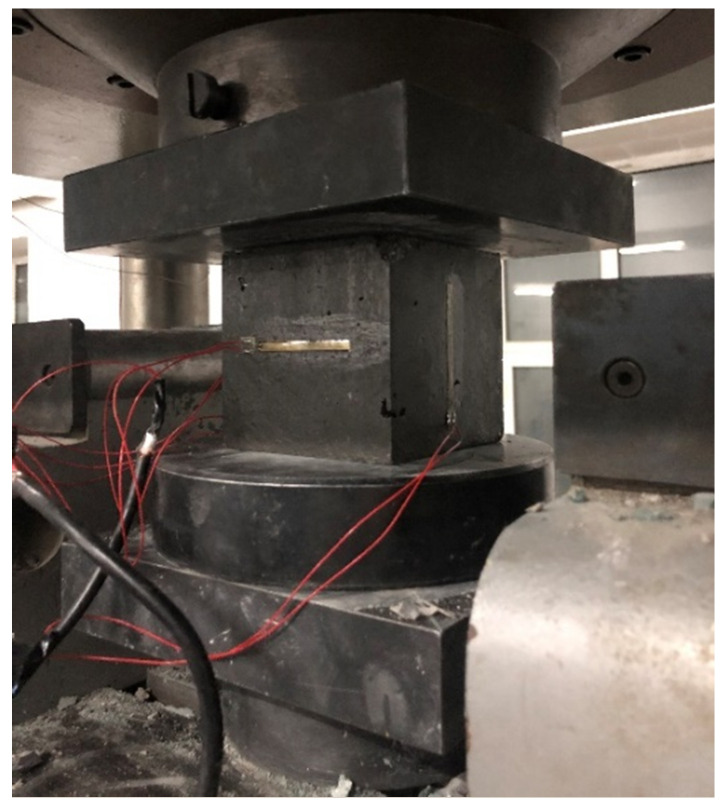
Compression test of EPUC.

**Figure 5 materials-14-03839-f005:**
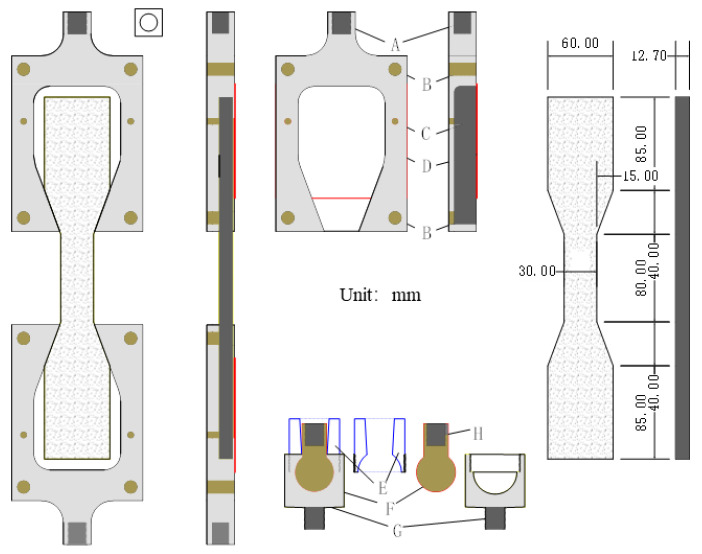
Dumbbell-shaped specimen and fixture size.

**Figure 6 materials-14-03839-f006:**
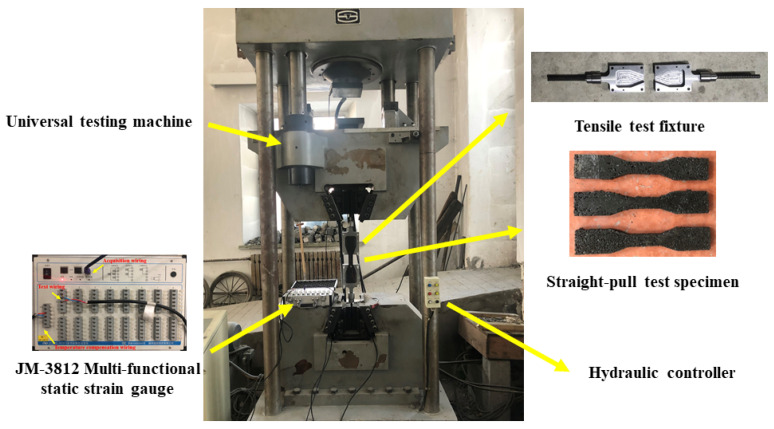
Straight tensile test of EPUC.

**Figure 7 materials-14-03839-f007:**
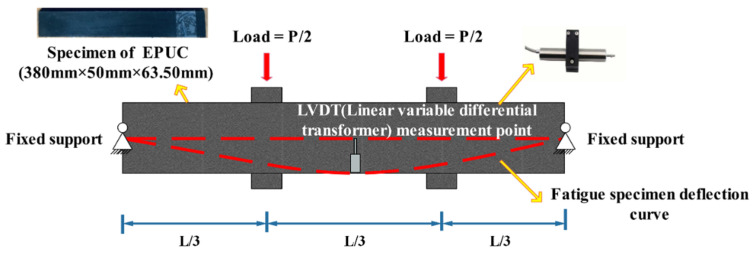
Loading diagram of four-point bending fatigue test.

**Figure 8 materials-14-03839-f008:**
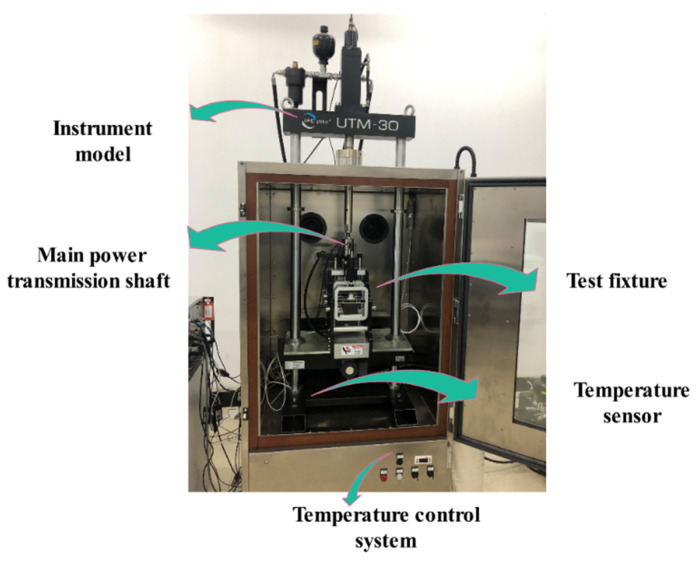
UTM-30 fatigue-testing machine.

**Figure 9 materials-14-03839-f009:**
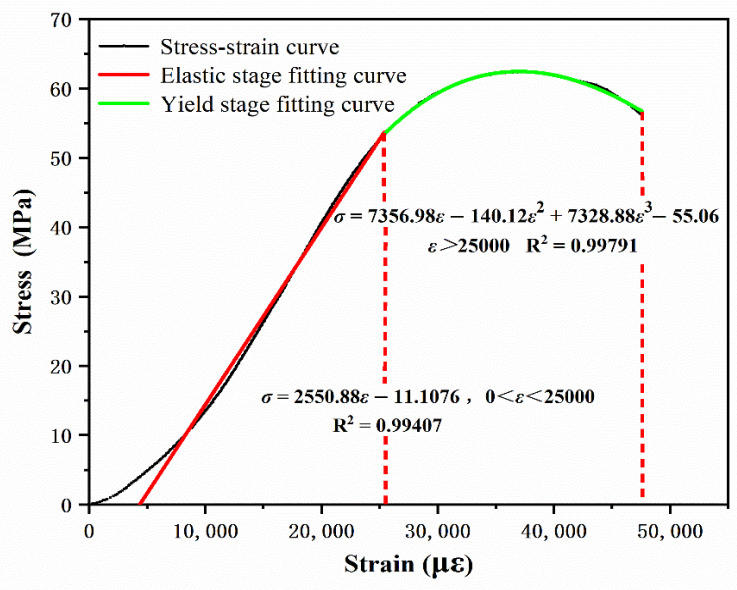
Compressive stress–strain curve diagram (average value).

**Figure 10 materials-14-03839-f010:**
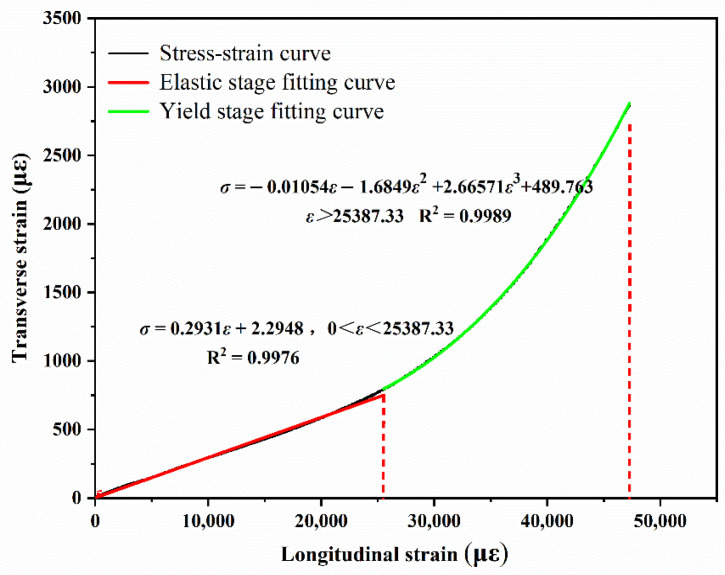
Bidirectional strain diagram (average value).

**Figure 11 materials-14-03839-f011:**
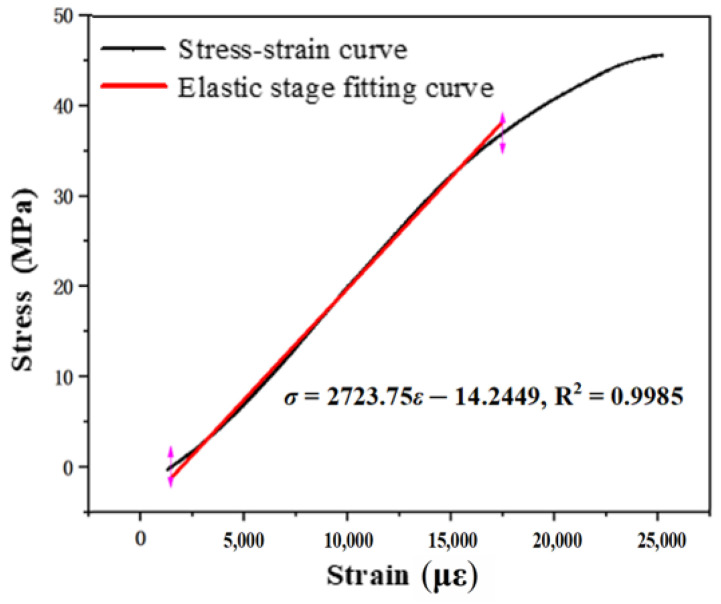
Stress–strain curve of direct tension.

**Figure 12 materials-14-03839-f012:**
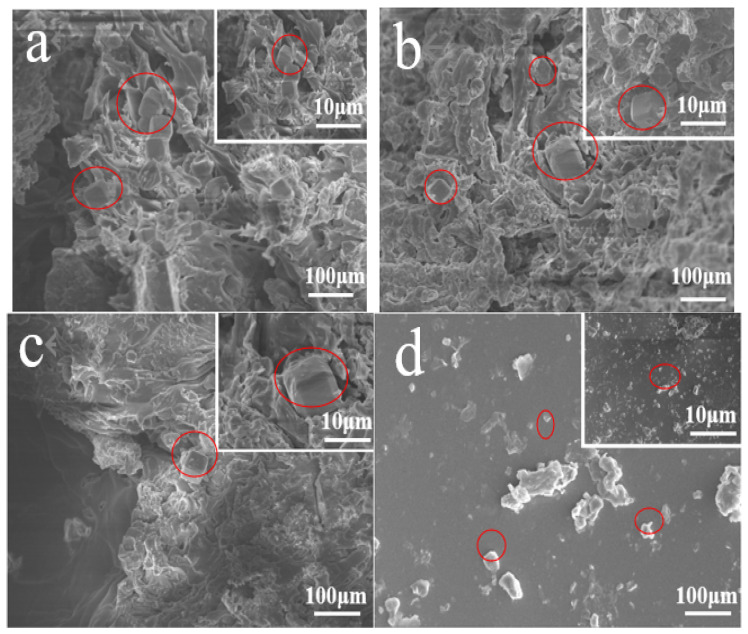
SEM images of EPUC ((**a**–**c**) are profiles, (**d**) is exterior view).

**Figure 13 materials-14-03839-f013:**
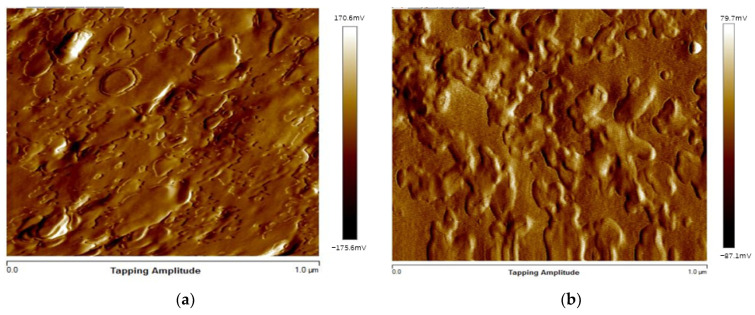
(**a**) Two-dimensional modeling diagram of AFM EPUC surface morphology; (**b**) 2D modeling diagram of AFM EPUC surface morphology.

**Figure 14 materials-14-03839-f014:**
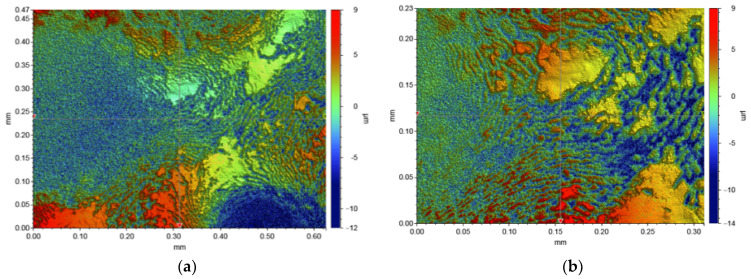
(**a**) Surface morphology of EPUC; (**b**) surface morphology of EPUC.

**Figure 15 materials-14-03839-f015:**
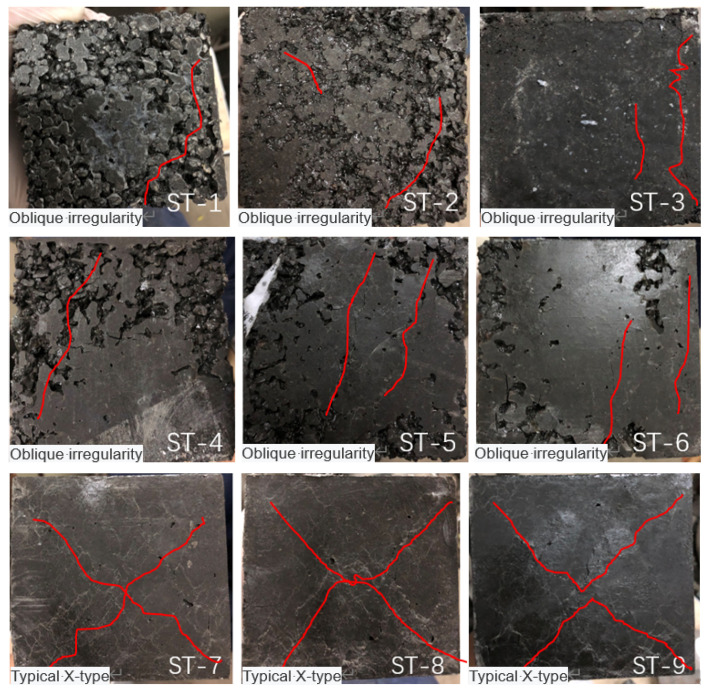
Axial compression failure patterns of specimens ST-1~ST-9.

**Figure 16 materials-14-03839-f016:**
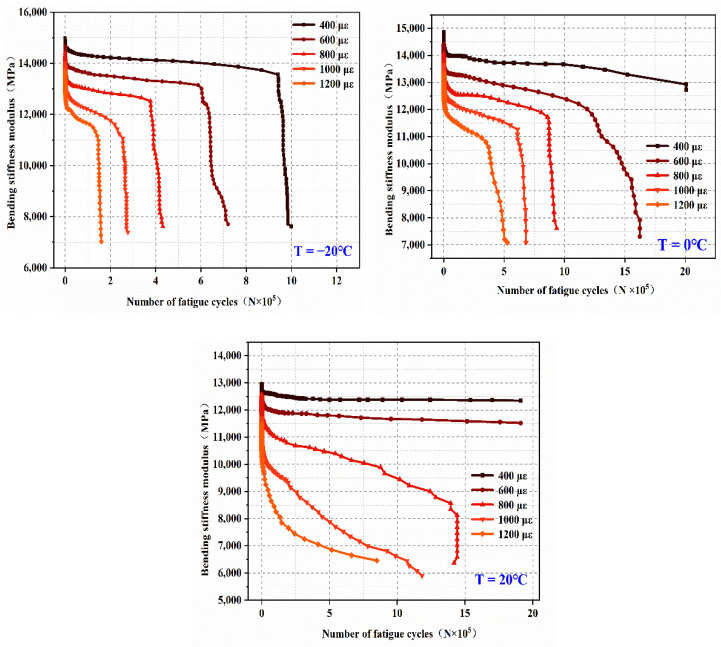
Attenuation rule of bending stiffness modulus.

**Figure 17 materials-14-03839-f017:**
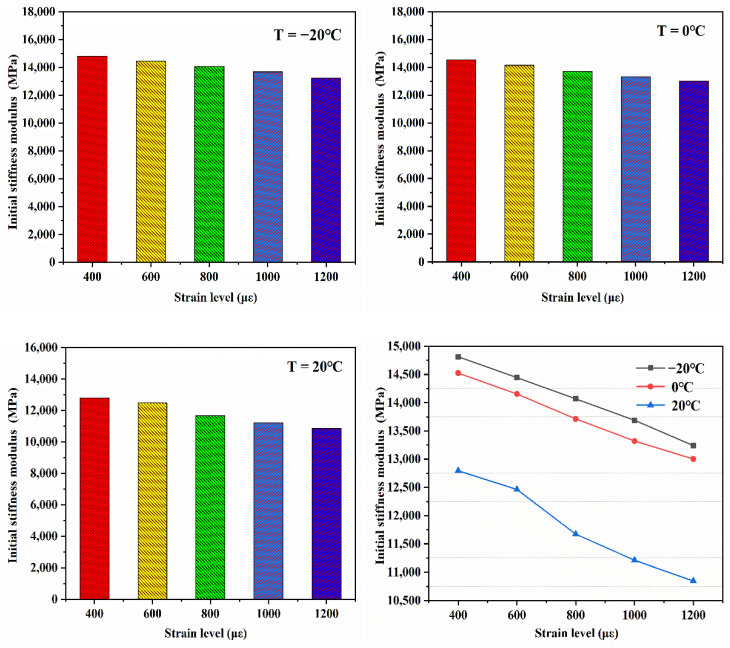
Test results of initial flexural stiffness modulus.

**Figure 18 materials-14-03839-f018:**
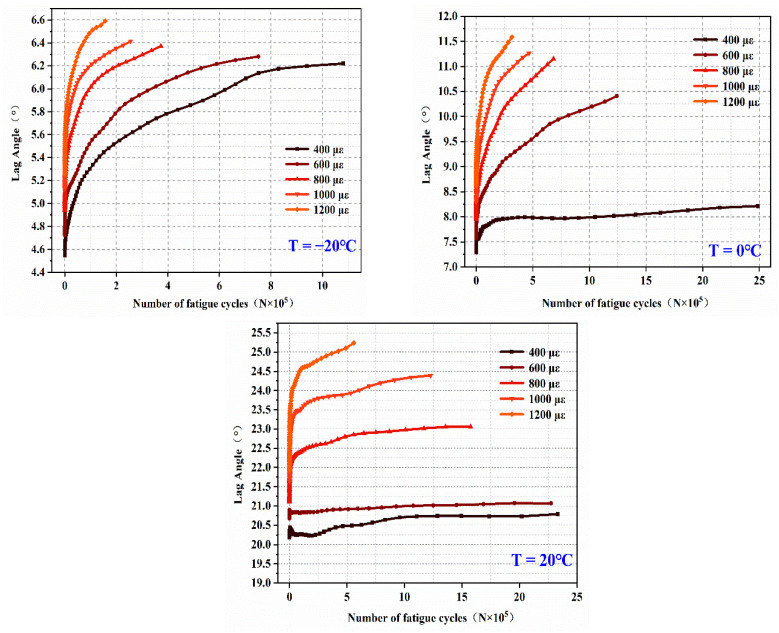
Variation of lag angle.

**Figure 19 materials-14-03839-f019:**
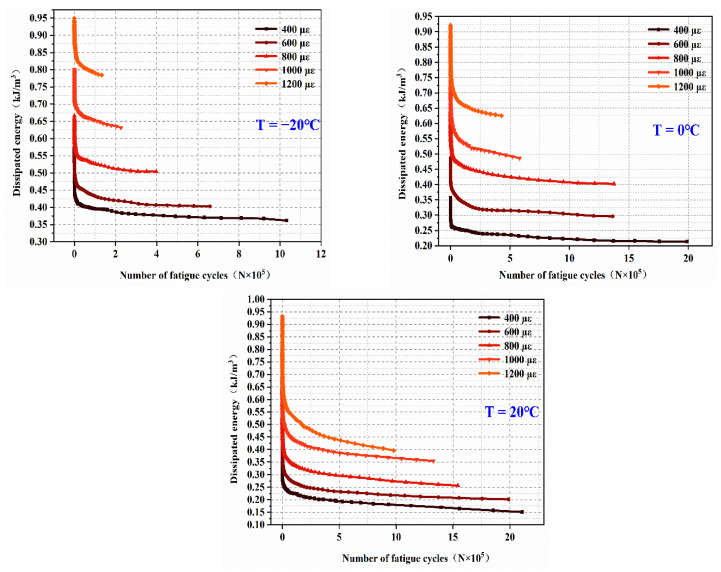
Dissipated energy loss under fatigue loading.

**Figure 20 materials-14-03839-f020:**
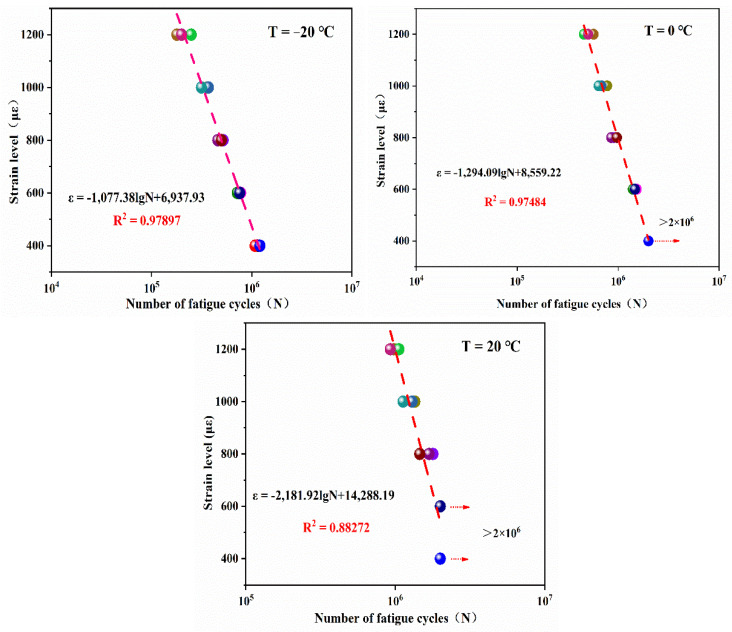
Fatigue life results and fitting curves at different temperatures.

**Figure 21 materials-14-03839-f021:**
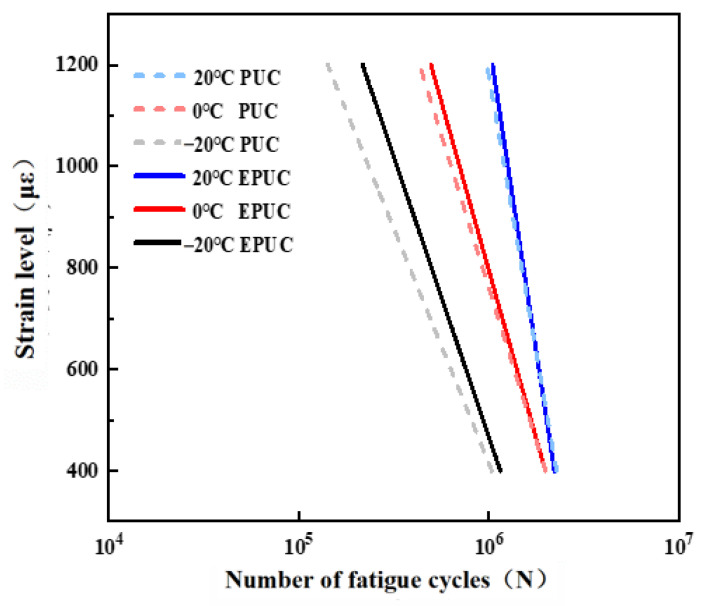
Comparison of fatigue life curves of EPUC and PUC at different temperatures.

**Table 1 materials-14-03839-t001:** Technical index of AC-10 fine-grained concrete.

Mesh Size (mm)	9.5	4.75	2.36	1.18	0.6	0.3	0.15	0.075	Mineral Powder
Percentage of through sieve	95.0%	60.0%	44.0%	32.0%	22.5%	16.0%	11.0%	6.0%	——
Aggregate mass (per 1000 g)	50.0	380.0	319.2	170.5	62.2	15.2	2.6	0.3	0.02

**Table 2 materials-14-03839-t002:** Compression strength values of EPUC.

Number	F01	F02	F03	F04	F05	F06	Average	Standard Deviation	Coefficient of Variation (%)
Density(kg/m^3^)	1760	1775	1774	1781	1759	1762	1768.5	86.7	4.902
Compression strength (MPa)	58.5	58.9	60.3	58.7	60.6	61.2	59.7	1.3	2.178

**Table 3 materials-14-03839-t003:** Tensile strength of EPUC.

Number	ZL01	ZL02	ZL03	ZL04	ZL05	ZL06	Average	Standard Deviation	Coefficient of Variation (%)
Density(kg/m^3^)	1760	1775	1774	1781	1759	1762	1768.5	86.7	4.902
Tensile strength (MPa)	40.9	41.9	40.3	41.6	40.9	41.1	41.1	0.322	0.783

**Table 4 materials-14-03839-t004:** Bending fatigue equation of polyurethane cement.

Number	Temperature	*S-N* Fatigue Equation
1	−20 °C	lgN = 7.4397 − 0.0672*σ*
2	0 °C	lgN = 7.6141 − 0.0559*σ*
3	20 °C	lgN = 7.5484 − 0.0332*σ*

## Data Availability

Not applicable.

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
