# Peer review of "Preparation and Mechanical-Fatigue Properties of Elastic Polyurethane Concrete Composites"

_materials, 2021, doi:10.3390/ma14143839_

Round 1
Reviewer 1 Report
The authors present a work on the elastic polyurethane concrete composites. The paper is interesting and of potential interest to readers. However, in my opinion, it still needs some improvements before being considered for publication.
In my opinion the abstract could be improved. The abstract must contain 5 basic information for this type of work, namely: should mention scope, motivation, methodology, results and expected impact of the research.
The first part of the introduction could perhaps explore a little more the advantages and disadvantages of this material and its potential interest for the proposed examples. The current state of knowledge presented in the introduction is limited, but sufficient to have a general idea about the work already carried out.
The differences between the six groups of specimens are not clear in the document. Are specimens from the same sample? In this case, it should present not only the mean, but also the standard deviation and the results variation coefficient.
In the following tests, it would be essential to also present some measure of results dispersion (for example, the standard deviation would already be sufficient).
What was mentioned in the previous points applied to all the properties studied.
In general, Chapter 3 presents and comments on the results obtained in an appropriate way. However, an important part of this type of work is missing: the comparison of the results obtained with values from other authors and/or reference values. Benchmarking is essential in this type of work.
The conclusions could be more assertive in recommending or limiting the use of this material.
Author Response
Dear Editor,
Thanks for the comments from you and the reviewers. We have carefully revised the manuscript according to the comments, and mark them in green. The responses are listed as follows.
Responds to the reviewer’s comments
Reviewer #1:
Comments and Suggestions for Authors
The authors present a work on the elastic polyurethane concrete composites. The paper is interesting and of potential interest to readers. However, in my opinion, it still needs some improvements before being considered for publication.
Comments1: In my opinion the abstract could be improved. The abstract must contain 5 basic information for this type of work, namely: should mention scope, motivation, methodology, results and expected impact of the research.
Response1:Thank you for your suggestions. We have revised and supplemented the abstract.
Comments2: The first part of the introduction could perhaps explore a little more the advantages and disadvantages of this material and its potential interest for the proposed examples. The current state of knowledge presented in the introduction is limited, but sufficient to have a general idea about the work already carried out.
Response2:Thank you for your suggestions. We have revised it, and marked them in green.
Comments3: The differences between the six groups of specimens are not clear in the document. Are specimens from the same sample? In this case, it should present not only the mean, but also the standard deviation and the results variation coefficient.
Comments4: In the following tests, it would be essential to also present some measure of results dispersion (for example, the standard deviation would already be sufficient). What was mentioned in the previous points applied to all the properties studied.
Response3 and 4:Thank you for your comments. We have supplemented the standard deviation and coefficient of variation in Tables 2 and 3.
Comments5: In general, Chapter 3 presents and comments on the results obtained in an appropriate way. However, an important part of this type of work is missing: the comparison of the results obtained with values from other authors and/or reference values. Benchmarking is essential in this type of work.
Response5:Thank you for your comments. We supplemented the tensile and compressive strength of EPUC and ordinary concrete materials for comparison, and added the comparison diagram of fatigue curves at different temperatures with EPUC conventional PUC materials, as shown in Fig. 21
Comments6: The conclusions could be more assertive in recommending or limiting the use of this material.
Response6:Thank you for your comments. We have rewrote the conclusion section.

Reviewer 2 Report
- please avoid general statements in the abstract ("a kind of", "certain amount") and present detailed information
- "Scanning electron microscope (SEM)" - why capital letter?
- 2.3. - it should be mechanical properties instead of mechanical property
- Fig. 4 - figures a and b have the same caption, why is it necessary to present both? What is the added value?
- Figures 6-9 are too basic - please check if they are all needed, in my opinion it would be better to combine them or remove some
- text on Figures 9-11 is not very visible, please correct it
- Figures 19, 21, 22 - remove x10^5 from each value on X axis and add it to the unit in the caption of X axis, unify all of the fonts used on graphs
- Figure 23 - N should be a number of fatigue cycles
- please complete and unify the literature according to the Journal's requirements
Author Response
Dear Editor,
Thanks for the comments from you and the reviewer2. We have carefully revised the manuscript according to the comments, and mark them in green. The responses are listed as follows.
Responds to the reviewer’s comments
Reviewer #2:
Comments and Suggestions for Authors
Comments1: please avoid general statements in the abstract ("a kind of", "certain amount") and present detailed information
Response1:Thank you for your suggestion. We have deleted "a certain amount of" in the abstract, and added the description of waste rubber particles "40 mesh with 10% fine aggregate volume replacement rate"
Comments2: "Scanning electron microscope (SEM)" - why capital letter?
Response2:Thank you for your comments. We have revised it.
Comments3: 2.3. - it should be mechanical properties instead of mechanical property.
Response3:Thank you for your comments. We have revised it.
Comments4: Fig. 4 - figures a and b have the same caption, why is it necessary to present both? What is the added value?
Response4:Thank you for your comments. Fig. 4(a) represents before the compression test and Fig. 4(b) represents after the compression test. There is no added value, so we have deleted Fig. 4(b).
Comments5: Figures 6-9 are too basic - please check if they are all needed, in my opinion it would be better to combine them or remove some
Response5:Thank you for your suggestion. we have combined Figures 6-9 into the new Figure 6.
Comments6: text on Figures 9-11 is not very visible, please correct it
Response6:Thank you for your suggestion. We have redrawn it Figures 9-11.
Comments7: Figures 19, 21, 22 - remove x10^5 from each value on X axis and add it to the unit in the caption of X axis, unify all of the fonts used on graphs
Response7:Thank you for your suggestion. We have redrawn Figure 19,21,22.
Comments8: Figure 23 - N should be a number of fatigue cycles
Response8:Thank you for your suggestion. We have redrawn Figure 21-23.
Comments9: please complete and unify the literature according to the Journal's requirements
Response9:Thank you for your suggestion. We revised the references according to the MDPI format.

Round 2
Reviewer 1 Report
The authors adequately answered all questions and changed the text according to the reviewers' recommendations.
In my opinion the paper presents conditions to be considered for publication.
Author Response
Dear Reviewer,
Thanks for the comments from you and the editor. We have carefully modified the manuscript according to the journal guidelines, and this paper was edited by a professional editor in MDPI. All changes are marked yellow.